# Photoacoustic/Ultrasound Endoscopic Imaging Reconstruction Algorithm Based on the Approximate Gaussian Acoustic Field

**DOI:** 10.3390/bios12070463

**Published:** 2022-06-27

**Authors:** Yongjun Wang, Chuqi Yuan, Jinsheng Jiang, Kuan Peng, Bo Wang

**Affiliations:** Department of Biomedical Engineering, School of Basic Medical Science, Central South University, Changsha 410017, China; 206512054@csu.edu.cn (Y.W.); 206511051@csu.edu.cn (C.Y.); 216512049@csu.edu.cn (J.J.)

**Keywords:** photoacoustic endoscopy, Gaussian acoustic field, dynamic focusing

## Abstract

This paper aims to propose a new photoacoustic/ultrasound endoscopic imaging reconstruction algorithm based on the approximate Gaussian acoustic field which significantly improves the resolution and signal-to-noise ratio (SNR) of the out-of-focus region. We demonstrated the method by numerical calculations and investigated the applicability of the algorithm in a chicken breast phantom. The validation was finally performed by the rabbit rectal endoscopy experiment. Simulation results show that the lateral resolution of the target point in the out-of-focus region can be well optimized with this new algorithm. Phantom experimental results show that the lateral resolution of the indocyanine green (ICG) tube in the photoacoustic image is reduced from 3.975 mm to 1.857 mm by using our new algorithm, which is a 52.3% improvement. Ultrasound images also show a significant improvement in lateral resolution. The results of the rabbit rectal endoscopy experiment prove that the algorithm we proposed is capable of providing higher-quality photoacoustic/ultrasound images. In conclusion, the algorithm enables fast acoustic resolution photoacoustic/ ultrasonic dynamic focusing and effectively improves the imaging quality of the system, which has significant guidance for the design of acoustic resolution photoacoustic/ultrasound endoscopy systems.

## 1. Introduction

Photoacoustic imaging (PAI) is an emerging imaging method that has flourished in recent years, featuring rich optical contrast, deep acoustic penetration depth, and high spatial resolution, enabling functional and molecular imaging [1,2,3]. However, due to the limitation of light penetration depth and the substantial attenuation of acoustic waves in biological organisms, it is challenging to visualize the deep-located internal biological tissues/organs [4]. Hence, to achieve this goal, photoacoustic endoscopy (PAE) has been developed over these years [5]. PAE has been employed as a valuable imaging modality in various applications, such as intravascular imaging [6,7,8,9], genitourinary imaging [10,11,12,13], esophageal [14,15,16], and gastrointestinal imaging [17,18,19,20,21].

PAE can be classified into acoustic resolution photoacoustic endoscopy (AR-PAE) and optical resolution photoacoustic endoscopy (OR-PAE). Among these, OR-PAE is characterized by high resolution yet shallow penetration depth, while AR-PAE is the exact opposite [22,23]. In particular, AR-PAE probes tend to use point-focused ultrasound transducers with the side-scanning mode. Through such probe structure and scanning mode, the system can simultaneously achieve acoustically focused photoacoustic/ultrasound endoscopic imaging because it allows two imaging modalities to share the same probe [24]. On this basis, a high-resolution and sensitive photoacoustic/ultrasound endoscopic imaging can be ultimately realized by employing an ultrasound transducer with a high numerical aperture.

Currently, a majority of acoustic resolution endoscopic studies are generally based on the B-mode method [25,26]. The major disadvantage of the B-mode method is that the acoustic field of the ultrasound transducer as well as the interaction between multiple ultrasound transducers are not considered during the image reconstruction process. Consequently, the lateral resolution and SNR of objects in the out-of-focus region are usually poor [27,28], which severely restricts the imaging depth. Therefore, a new image reconstruction algorithm is urgently needed to achieve dynamic focusing of objects in AR-PAE, even at different depths.

Compared to B-mode, in acoustic resolution photoacoustic microscopy (AR-PAM), the lateral resolution and SNR in the out-of-focus region can be improved by the synthetic aperture focusing technique (SAFT) coupled with the coherence factor method [29,30,31,32]. However, this algorithm only performs data summing within the numerical aperture of the transducer, in the application of AR-PAE, the limited number of A-lines used for summation leads to high artifacts, which severely reduces the SNR of the obtained images [33]. In addition, the coherence factor used in the algorithm is nonlinear and susceptible to noise disturbance, which is not conducive to improving the reliability of the image. More recently, the improved back-projection (BP) algorithm has been used in AR-PAE [27,28,33], which effectively improves the resolution and SNR in the out-of-focus region. Yet compared with SAFT, this method is more computationally intensive, and the computational speed is very slow in 3D imaging, severely limiting its application.

In this study, a new acoustically focused photoacoustic/ultrasound endoscopic imaging algorithm based on the approximate Gaussian focused acoustic field, called the Gaussian-beam-based back-projection (GB-BP) method, is proposed to significantly improve the resolution and SNR in the out-of-focus region. We demonstrated the method by numerical calculations and investigated the applicability of the algorithm in a chicken breast phantom. The validation was finally performed by the rabbit rectal endoscopy experiment.

## 2. Methods

### 2.1. Reconstruction Algorithms

Figure 1 shows the scheme of the acoustical focused photoacoustic/ultrasound endoscopic imaging algorithm. Firstly, according to the focused acoustic field of the ultrasound transducer, a suitable equivalent focal point is found, the distance F from the transducer to the focal point is calculated, as well as the equivalent beam waist radius $\omega_{0}$ and equivalent semi-focal depth $z_{0}$ corresponding to the acoustic field of the ultrasound transducer can be defined.

With the found parameters above, the final pixel value of an individual pixel point can be calculated according to the following delay superposition formula:(1)$$I(\vec{r})=\sum_{i=1}^{Nd}A(i,\vec{r})\times S(i,\Delta t(i,\vec{r}))\;.$$

Here, Nd is the number of directional positions of the transducer scan, $\vec{r}$ is the coordinate of the pixel to be reconstructed, S(i,t) represents the photoacoustic signal received by the transducer at the position i, and t is the time. Δt and A are the phase and weight coefficients, respectively, which are determined by the position of the required pixel in the approximate Gaussian acoustic field, that is, the position of the pixel relative to the transducer.

Where, in photoacoustic imaging, the phase coefficient can be written as follows:(2)$$\Delta t(i,\vec{r})=\frac{F+a\sqrt{1+\frac{b^{2}}{a^{2}+z_{0}^{2}}}}{v}.$$

While in ultrasound imaging, it becomes:(3)$$\Delta t(i,\vec{r})=\frac{F+a\sqrt{1+\frac{b^{2}}{a^{2}+z_{0}^{2}}}}{\frac{v}{2}},$$
where v is the propagation velocity of ultrasound in the medium, and the parameters a and b are the axial and radial distances of the pixel concerning the focal point of the transducer, respectively, which can be obtained as follows:(4)$$a=\frac{\left({\vec{F}}_{i}-{\vec{O}}_{i})\cdot (\vec{r}-{\vec{F}}_{i}\right)}{\left|{\vec{F}}_{i}-{\vec{O}}_{i}\right|}$$
(5)$$b=\left|(\vec{r}-{\vec{F}}_{i})-\frac{({\vec{F}}_{i}-{\vec{O}}_{i})\cdot a}{\left|{\vec{F}}_{i}-{\vec{O}}_{i}\right|}\right|.$$

For this equation, ${\vec{F}}_{i}$ and ${\vec{O}}_{i}$ are the focus and the center of rotation at the i position of the transducer, respectively.

To obtain the final image, the other mentioned weight coefficient A in the approximate Gaussian acoustic field can be calculated by the following formula:(6)$$A(i,\vec{r})=\left(\sqrt{\frac{z_{0}^{2}}{z_{0}^{2}+a^{2}}}\cdot e^{\frac{-{\left(bz_{0}\right)}^{2}}{2w_{0}^{2}(a^{2}+z_{0}^{2})}}\right)\times f(i,\vec{r}),$$
where $f(i,\vec{r})$ refers specifically to a position-limiting function determined by the current pixel relative to the transducer position, which represents the range of the delayed superposition of the acoustic field in the following way:(7)$$f(i,\vec{r})=\left\{ \begin{array}{cc} 1 & a_{1}\leq a\leq a_{2},b_{1}\leq b\leq b_{2}\\ 0 & else \end{array}\right..$$

The parameters $a_{1}$, $a_{2}$, $b_{1}$ and $b_{2}$ are the four parameters of the position restriction function, which can be calculated by the following equations:(8)$$a_{1}=L-F$$
(9)$$a_{2}=R-F+L$$
(10)$$b_{1}=-R$$
(11)$$b_{2}=R.$$

Here, R is the radius of the imaging area, and L is the distance from the virtual transducer to the center of rotation.

### 2.2. System Installations

The configuration of the PA/USE experimental system used is illustrated in Figure 2. The pulsed excitation light was emitted by a 532 nm pumped optical parametric oscillator (OPO) laser (SpitLight OPO 600 mid-band, InnoLas, Krailling, Germany) with a repetition frequency of 20 Hz and a tunable range of 680 nm to 1320 nm. The laser beam was shaped by a combination of flat-convex and flat-concave lenses, entered the water tank through the diaphragm, and finally was the incident in the front of the probe through a 45-degree calcium fluoride lens. The length of the probe shell was approximately 15 cm, and an aluminum off-axis parabolic mirror (#37-282, EdmundOptics, Barrington, NJ, USA) was mounted at the distal end for both the reflection of the laser and the acquired ultrasound signal. The optical density at the tissue surface was kept below 20 mJ/cm^2^, which was within the American National Standards Institute (ANSI) safety limit [34]. The acquired ultrasound was reflected through a 45-degree calcium fluoride lens and eventually received by a 10 MHz flat-field ultrasound transducer.

The probe shell was made of a transparent polyethylene tube with an outer diameter of 8 mm, a wall thickness of 0.5 mm, and a parabolic mirror with a diameter of 6.25 mm. The probe was driven by a circular scanning stepper motor with a belt gear. The received photoacoustic signal was amplified by an RF amplifier (DPR500 Pulser-Receivers, Imaginant, Pittsford, NY, USA), digitized (NI PCI-5124, 200 MS/s, 12-Bit, National Instruments Corporation, Austin, TX, USA), and then stored on a computer host for further processing. In photoacoustic imaging mode, the entire system was synchronized by a laser. The laser was turned off during the ultrasonic endoscopic (USE) imaging mode, and an RF amplifier was used as the ultrasound transmitter. The feedback ultrasound signal was received by the transducer and then amplified by the RF amplifier and stored on the computer.

### 2.3. Numerical Simulations

In the simulation, there are nine point-targets uniformly located between 3~11 mm on the x-axis. As shown in Figure 3, the black solid dots show the distribution of targets and their corresponding locations. The center of rotation of the probe is set to be the origin (0, 0). The focused ultrasound transducer had a bandwidth of 60% with a diameter of 5 mm. The target was scanned in two dimensions at a radius of 12.5 mm, with the center of the transducer at the origin, and the scanning angle range was 30 degrees with an angular interval of 0.25 degrees. During the simulation, the center frequency of the transducer was 10 MHz, with a focal length of 7 mm and a numerical aperture of 0.5054, and the sampling frequency was selected to be 200 MHz.

As shown in Figure 3, the black dotted box is the range of image reconstruction. Both the proposed PA/USE imaging algorithm and the B-mode method were used to calculate and process each set of data, and the reconstruction results of the two imaging methods were analyzed separately from each other. The lateral full width at half maximum (FWHM) of the simulated target was extracted as its lateral resolution, respectively and 200 trials were used to simulate the influence of random noise on the signal. In each trial, Gaussian white noise was added to the data by using the Box–Muller method at a standard deviation of 5% of the maximum signal amplitude. The average of the 200 reconstructed amplitudes of each target was taken as the signal S, and the standard deviation of these reconstructed amplitudes was taken as the noise N. The SNR was calculated as:(12)$$SNR=20lg\frac{S}{N}.$$

In addition to the above-mentioned simulation and data analysis of the axial target located on the X-axis, we also simulated the targets distributed in the lateral direction. As shown in Figure 3, the red circle shows the targets distribution in the simulation experiment. The red solid line box is the imaging area, and the parameter settings are consistent with the above simulation experiment. We used the GB-BP algorithm and the B-mode method to reconstruct the simulated data respectively, and analyzed the lateral resolution of the targets.

All the simulations above were performed by MATLAB.

### 2.4. Phantom Experiments

In the phantom experiment, the chicken breast was used to build a phantom that has a similar optical scatter coefficient to real biological tissue.

To examine the optimized capability of the proposed algorithm for imaging targets at large depths, two PVC tubes were placed under the chicken breast tissue and imaged at 805 nm. The PVC tubes both had an inner diameter of 1 mm and an outer diameter of 2 mm. For one tube, it was filled with 0.5 mg/mL ICG in 22.5% albumin, and the other tube was only filled with albumin solution. Both tubes were buried at a depth of 18 mm under the chicken breast meat tissue.

To determine the SNR of both tubes at each depth, a small background region was selected at approximately the same depth with the standard variation in this region used as the noise and the peak signal of each tube used as the signal amplitude. Meantime, we extracted the lateral profile of two tubes in the reconstructed image and calculated the FWHM.

### 2.5. Rabbit Rectal Endoscopy Experiment

We performed a rabbit rectal endoscopic imaging experiment using the mentioned laboratory apparatus to verify the effectiveness of the proposed algorithm of in vivo imaging.

Photoacoustic and ultrasound imaging was performed on a male New Zealand white rabbit, weighing approximately 2 kg, which was fasted for 24 h before the experiment. Each rabbit was first given a rectal enema with saline, then fixed on a steel frame, where the probe was inserted into the rabbit’s rectum at a depth of approximately 6 cm, and the photoacoustic images, as well as ultrasound images, were acquired at the same location. The scanning angle range was 360 degrees with an angular interval of 0.9 degrees. During all imaging experiments, gas anesthesia was performed using isoflurane at a dose of 2% with a flow rate of approximately 1 L/min. At the end of the experiments, all rabbits survived in good condition.

The image reconstruction was performed using the algorithm proposed and the B-mode method. All animal experimental procedures were approved by the Department of Laboratory Zoology, Central South University (No. 2020KT-39).

## 3. Results

### 3.1. Numerical Simulations

The results of the numerical simulation are shown in Figure 4. Figure 4a shows the photoacoustic images reconstructed by using the B-mode method and the GB-BP algorithm, respectively, with the lateral resolution and SNR calculated for the reconstructed results of both algorithms. Figure 4b shows the variation of lateral FWHM and SNR with imaging depth for the targets obtained after processing by both algorithms. Compared with the conventional B-mode method, the SNR of the image obtained by processing with the GB-BP algorithm is significantly improved, from 25 dB to 35 dB at the focal position of 7 mm. When the conventional B-mode method is used for photoacoustic imaging, the best lateral resolution can only be achieved at the focal point of acoustic focus. Instead, by using the GB-BP algorithm, the lateral resolution of the target point within the unfocused region can be effectively improved, and the lateral resolution gradually increases with distance. Figure 4c,d, on the other hand, show the ultrasound images obtained by these two methods as well as the lateral resolution and SNR. Similar to the experimental results of photoacoustic imaging, the SNR is significantly improved with the GB-BP algorithm, where the SNR is improved from 16 dB to 36 dB at the near 3 mm position. Among the reconstruction results of the GB-BP algorithm, the lateral resolution of the target point in the out-of-focus region is well optimized, with the lateral resolution improved from 1.48 mm to 0.11 mm at the position of 3 mm in the near distance.

Figure 5a,b show the photoacoustic and ultrasound images reconstructed by using the B-mode method and the GB-BP algorithm, respectively, including the lateral target points. Comparing the reconstruction results of the two algorithms, except that the lateral resolution of the targets near the focus is slightly worse, the targets in the reconstruction result of our proposed algorithm have a smaller lateral resolution. This is consistent with the above simulation results. We calculated the lateral resolution of targets at Y = 2 and Y = 0 in the photoacoustic and ultrasound images, respectively, as shown in Figure 5c,d. It can be seen that the lateral resolution at Y = 0 is better than Y = 2. In the photoacoustic image, the lateral resolution of the target at (3, 0) is 0.1875 mm, which is better than 0.2275 mm at (3, 0). As the imaging depth increases, the distances of the targets at Y = 2 and Y = 0 relative to the origin become approximately the same, and the lateral resolution gradually approaches. Ultrasound results also show this trend.

### 3.2. Chicken Breast Phantom Experiment

Results of the chicken breast phantom experiments are shown in Figure 6. Figure 6a shows the actual images of the two PVC tubes in the chicken breast phantom, and tube 2 (right) contains ICG. Figure 6b displays the photoacoustic and ultrasonic images reconstructed respectively by two algorithms when the PVC tube is placed 18 mm deep below the surface of the chicken breast tissue. The reconstruction time of a single image using the GB-BP algorithm is 8 s. It is noteworthy that both tubes can be visualized in the ultrasound image, while only the PVC tube with ICG solution can be shown in the photoacoustic image, which indicates the specificity of photoacoustic imaging. Table 1 provides the SNR of the two tubes calculated by the B-mode method and GB-BP algorithm. Since tube 1 only contains albumin solution, it is not shown in the photoacoustic image, the SNR of tube 1 in the photoacoustic image is not calculated here. It can be seen that the SNR of tube 2 in the photoacoustic image obtained by the GB-BP algorithm is about 5.4 dB higher than the result of the B-mode method. The PVC tube with ICG can still be well distinguished at a depth of 18 mm. Specifically, the SNRs were 48.0 dB and 39.2 dB for the two PVC tubes after processing with the GB-BP algorithm, which was improved by 6.6 dB and 3.4 dB, respectively. Table 2 shows the calculated FWHMs of the lateral profiles for two target tubes and the improvements of the calculated FWHMs with our GB-BP algorithm. The FWHM data for tube 1 in the PA image is also not shown here. From this table, it can be seen that the FWHM of the two tubes in the image can be significantly reduced by using the GB-BP algorithm in both the photoacoustic and the ultrasound image. The lateral resolution of tube 2 in the PA image was reduced from 3.975 mm to 1.857 mm by using our new algorithm, which is a 52.3% improvement. In addition, in the ultrasound results, the lateral resolution of tube 1 is improved by 37.7%, and the improvement for tube 2 with our new method is about 32.7%.

### 3.3. Rabbit Rectal Endoscopy Experiment

The picture of the rabbit rectal endoscopy experiment is shown in Figure 7a. Figure 7b presents the rabbit rectal photoacoustic endoscopy images obtained using the B-mode method and the GB-BP algorithm, respectively. The images acquired with the GB-BP algorithm generally have better image quality compared to the B-mode method. At the white arrows, there is a clearer and more coherent structure of the rabbit rectum by using the GB-BP algorithm.

The endorectal ultrasound images of rabbits processed by the B-mode method and GB-BP algorithm are shown in Figure 7c, respectively. Similarly, the GB-BP algorithm provides a much higher image quality. The area outlined by the white ellipse is the hypoechoic region near the inner rectal wall, which appears as a narrow-shadowed area. Compared with the B-mode method, in the ultrasound endoscopic image obtained by using the GB-BP algorithm, the boundary contained in the solid red box is clearer and more coherent.

To further demonstrate the depth imaging advantage of the proposed algorithm, we injected 0.5 mL ICG (2.5 mg/mL) into the peripheral tissue on the left side of the rectum of another rabbit and compared the photoacoustic reconstructed images of the B-mode method and the GB-BP algorithm, as shown in Figure 8. The results clearly show that the photoacoustic signal of ICG (green elliptical region) can be well distinguished at a distance of about 6 mm from the probe surface. In addition to the ICG photoacoustic signal, several visceral structures located at a depth of 15 mm can also be visualized. All these results above prove that the GB-BP algorithm is capable of providing much higher quality photoacoustic images.

## 4. Discussion

The dynamic focus of acoustically focused photoacoustic/ultrasound endoscopic imaging is significantly related to image quality. Since high-frequency ultrasonic transducers are generally used in acoustically focused photoacoustic/ultrasonic imaging, the resolution of the out-of-focus region will be poor [35]. In this paper, we propose the Gaussian-beam-based back-projection algorithm as a way to achieve fast acoustically focused photoacoustic/ultrasound endoscopic imaging reconstruction. We performed simulations and experimental validation of our method and demonstrated that our method can achieve fast dynamic focusing.

For our proposed imaging algorithm, we examined its performance. We found that:Our algorithm improves the lateral resolution of the out-of-focus region, and the lateral resolution gradually increases with distance. This is due to the effect of dynamic focusing, and the relative numerical aperture of the distant target becomes smaller, so the resolution becomes worse;Results show that, under the same central frequency, the lateral resolution of AR-USE is higher than that of the AR-PAE. This is because ultrasound imaging is a double-path (pulse-echo) response process, so its spatial impulse response is a convolution of the transducer’s forward and backward responses. In contrast, the spatial impulse response in photoacoustic imaging is only the transducer’s backward response, so the FWHM of the resulted lateral profile is wider than that in AR-USE.After dynamic focusing of ultrasound, the signal intensity of the point target in the out-of-focus region becomes worse, which is attributed to the lower intensity of the emitted acoustic field in the out-of-focus region.

The chicken breast phantom experiments indicate that the GB-BP algorithm significantly improves the lateral resolution of the targets in the photoacoustic/ultrasound images, and the improvement is much greater in photoacoustic imaging than that in US imaging, which reaches 52.3%. It is the opposite of the previous simulation results, where ultrasound imaging results do not show better lateral resolution improvement. We think there may be two reasons. On the one hand, the contours of the targets in ultrasound and photoacoustic imaging are inconsistent, and the contours of the objects in the ultrasound images are most likely the reconstruction results of the ultrasound signals reflected from the outer wall of the PVC tube. The lateral resolution of the objects in the ultrasound image is close to 2 mm, which is almost the same as the outer diameter of the tube. However, the target in the photoacoustic image is the ICG solution confined within the diameter of 1 mm, so the lateral resolution in the photoacoustic image is better. On the other hand, the transmission of ultrasound in real biological tissue is affected by the refraction and reflection which are caused by the inhomogeneous acoustic parameters. The length of the acoustic transmission in the US is twice as long as that in PA, which makes the propagation of ultrasound further deviate from the idea linear model, and introduces more model error to the reconstruction results.

The results of the rabbit rectal endoscopy experiment demonstrate that the improvement of image quality is limited, and some boundaries are still incoherent, which does not achieve the effect in the simulation experiment. This may be because we use point targets in simulation, but the imaging targets in vivo are more complicated. In future work, we need to further improve our algorithm to deal with the targets with complex shapes. Furthermore, the motion artifact caused by the peristalsis of the rectum is also an important reason to limit the improvement of image quality. This can be solved by using a micro acoustic array which can acquire the acoustic signal from the imaging area instantaneously.

In addition to enabling dynamic focusing of photoacoustic/ultrasound imaging, compared with the commonly SAFT algorithm in AR-PAM, the main advantage of our algorithm is that it does not consider whether the pixel is inside or outside the conical receiving area, thus allowing uniformity in programming and simplifying the programming structure. In the meantime, the reconstruction time of a single image using the GB-BP algorithm is 8 s, which is much faster than the Improve BP algorithm [27]. However, in our algorithm, the algorithm parameters have an impact on the imaging effect, such as $\omega_{0}$ and $z_{0}$. We need to develop a more accurate model-based dynamic focusing algorithm to further improve the robustness. The signal intensity in the out-of-focus region cannot be significantly improved in ultrasound imaging even with the dynamic focusing algorithm, therefore we may consider using a long-focus depth ultrasound transducer with a more homogeneous acoustic field to ease the system configuration.

## 5. Conclusions

In this work, we propose a fast photoacoustic/ultrasound endoscopic imaging reconstruction algorithm based on the approximate Gaussian acoustic field of the focused transducer, which can effectively realize dynamic focusing and expand the imaging depth of the field of the system. We have performed simulation and phantom experiments to validate the algorithm. Numerical simulations indicate that the algorithm can improve lateral resolution from 1.48 mm to 0.11 mm and SNR from 16 dB to 36 dB for objects at 3 mm. Meanwhile, phantom experimental results show that the lateral resolution of the ICG tube in the photoacoustic image is reduced from 3.975 mm to 1.857 mm by using our new algorithm, which is a 52.3% improvement. Ultrasound images also show a significant improvement in lateral resolution. Moreover, we perform in vivo experimental validation of the algorithm. The results of the rabbit rectal endoscopy experiment prove that the algorithm we proposed is capable of providing higher-quality photoacoustic/ultrasound images with a reconstruction speed of 8 s/frame.

In conclusion, the algorithm enables fast acoustically focused photoacoustic/ultrasonic dynamic focusing and effectively improves the imaging quality of the system, which has significant guidance for the design of acoustically focused ultrasound/photoacoustic endoscopy systems.

## Figures and Tables

**Figure 1 biosensors-12-00463-f001:**
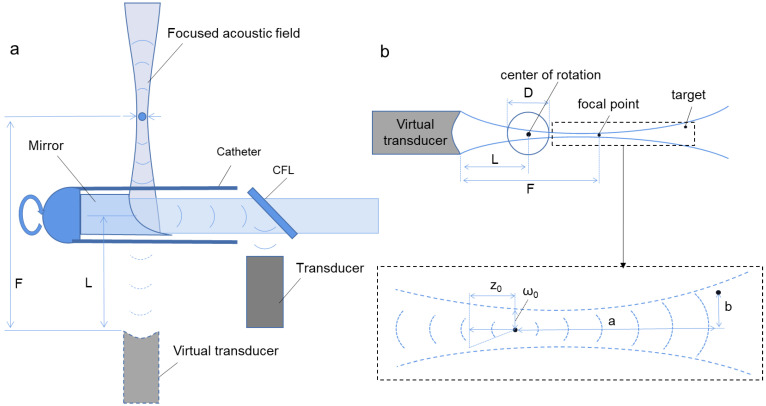
The schematics of the acoustically focused photoacoustic/ultrasound endoscopic imaging algorithm. (**a**) The diagram of the probe. Using a parabolic mirror, which acts as an acoustic lens, creates a focused acoustic field at the side of the probe; (**b**) The schematic of the approximate Gaussian acoustic field. The parabolic mirror rotates, i.e., the virtual transducer rotates with radius L. CFL: Calcium Fluoride Lenses. D is the diameter of the photoacoustic endoscopic probe.

**Figure 2 biosensors-12-00463-f002:**
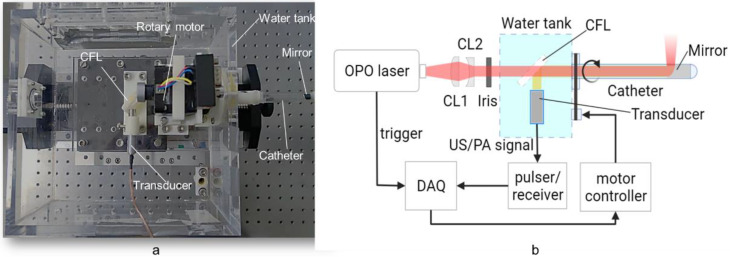
(**a**) Photo of the front end of the endoscopic system; (**b**) Schematic of the experimental PAE setup. CFL: Calcium Fluoride Lens; CL1: flat-convex lens; CL2: flat-concave lens; DAQ: data acquisition card.

**Figure 3 biosensors-12-00463-f003:**
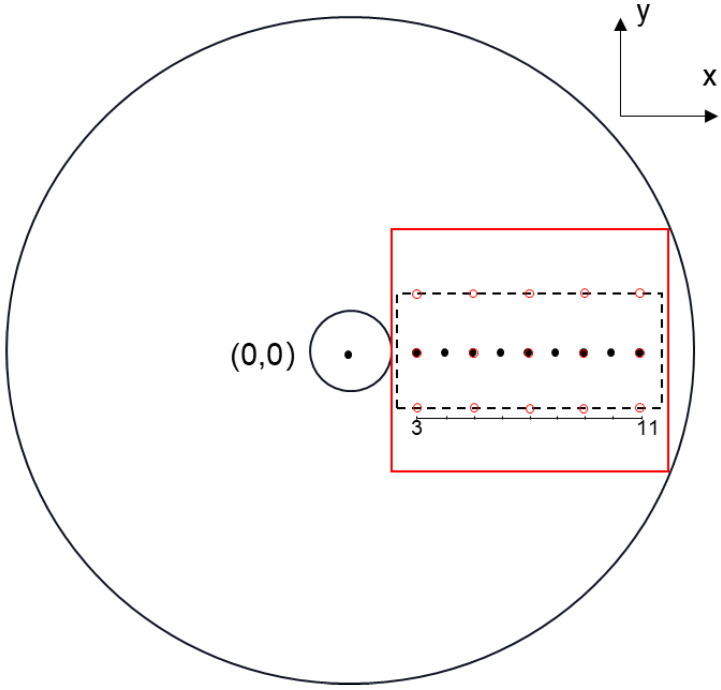
Schematic of numerical simulation. Coordinate axis units are millimeters.

**Figure 4 biosensors-12-00463-f004:**
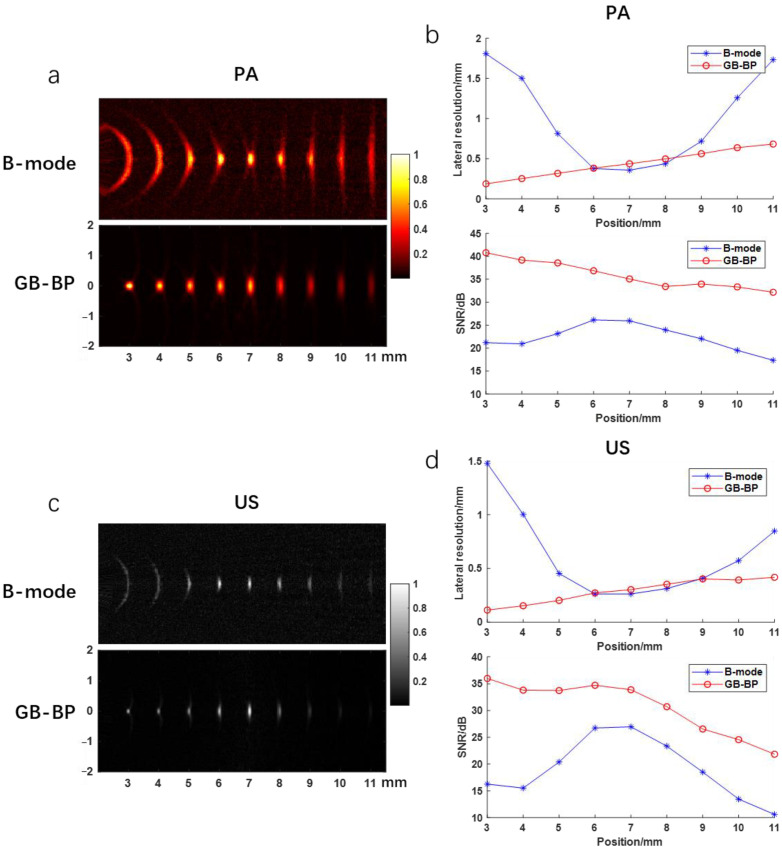
(**a**) The photoacoustic images reconstructed by the B-mode method and GB-BP algorithm; (**b**) The lateral resolution and SNR of the photoacoustic results in (**a**); (**c**) The ultrasonic images reconstructed by the B-mode method and GB-BP algorithm; (**d**) The lateral resolution and SNR of the ultrasonic results in (**c**). Coordinate axis units are millimeters.

**Figure 5 biosensors-12-00463-f005:**
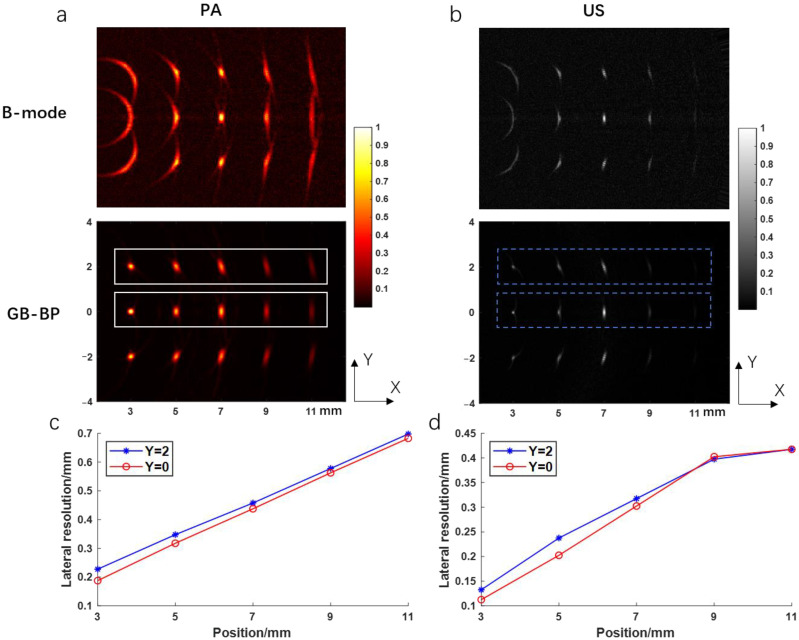
(**a**) The photoacoustic images reconstructed by the B-mode method and GB-BP algorithm; (**b**) The ultrasonic images reconstructed by the B-mode method and GB-BP algorithm; (**c**) Lateral resolution of the targets in the white solid line box in (**a**); (**d**) Lateral resolution of the targets in the blue dashed box in (**b**). Coordinate axis units are millimeters.

**Figure 6 biosensors-12-00463-f006:**
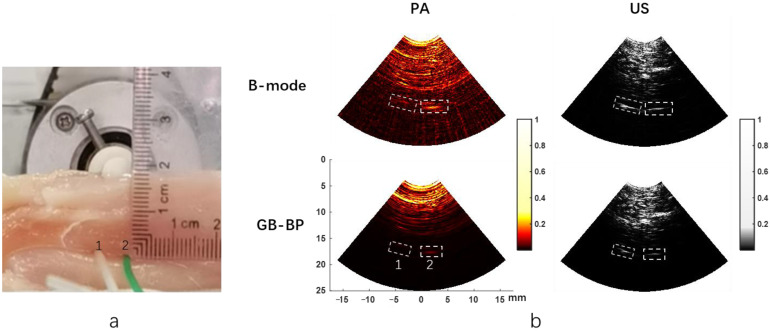
(**a**) The picture of chicken breast phantom experiments; (**b**) The photoacoustic and ultrasonic images reconstructed by two algorithms when the PVC tube is placed 18 mm deep below the surface of the chicken breast tissue. Two white dashed boxes contain two PVC tubes, and right tube 2 contains the ICG. Coordinate axis units are millimeters.

**Figure 7 biosensors-12-00463-f007:**
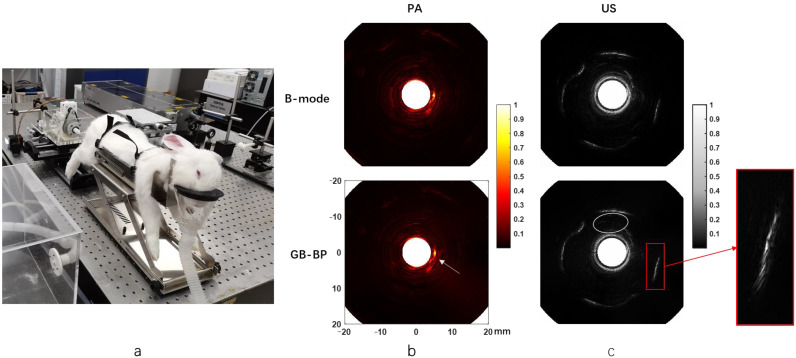
(**a**) The picture of the rabbit rectal endoscopy experiment; (**b**) The rabbit rectal photoacoustic endoscopy images obtained by the B-mode method and GB-BP algorithm; (**c**) The endorectal ultrasonic images of rabbits processed by the B-mode method and GB-BP algorithm. Coordinate axis units are millimeters.

**Figure 8 biosensors-12-00463-f008:**
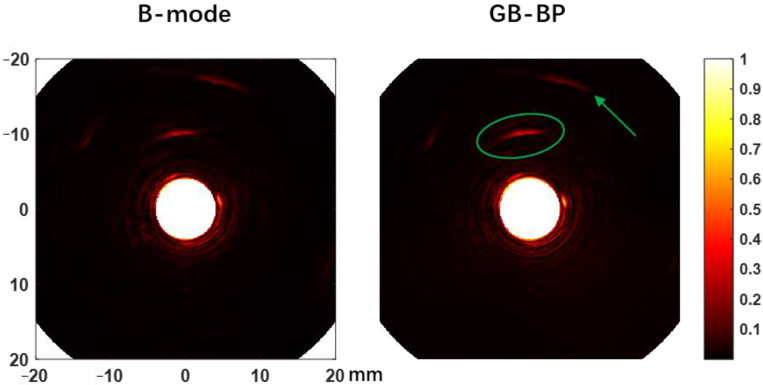
The rabbit rectal photoacoustic endoscopy images after ICG injection obtained by the B-mode method and GB-BP algorithm. Coordinate axis units are millimeters.

**Table 1 biosensors-12-00463-t001:** The calculated SNRs of two targets with different reconstruction algorithms.

Target Index	SNRs (dB) in PA Image		SNRs (dB) in US Image	
	B-Mode	GB-BP	B-Mode	GB-BP
1	/	/	41.4	48.0
2	21.9	27.3	35.8	39.2

**Table 2 biosensors-12-00463-t002:** The extracted FWHMs of the lateral profiles for two targets.

Target Index	PA				US			
	Calculated FWHM (mm)		Improvement		Calculated FWHM (mm)		Improvement	
	B-mode	GB-BP	In mm	In %	B-mode	GB-BP	In mm	In %
1	/	/	/	/	3.462	2.157	1.305	37.7
2	3.975	1.857	2.118	52.3	3.353	2.257	1.096	32.7

## Data Availability

The data presented in this study are available in this article.

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
