# Peer review of "Photoacoustic/Ultrasound Endoscopic Imaging Reconstruction Algorithm Based on the Approximate Gaussian Acoustic Field"

_biosensors, 2022, doi:10.3390/bios12070463_

Round 1

Reviewer 1 Report

Dear Authors,

I found your manuscript very interesting due to the attempt to exceed the imaging resolution, the development of the method and experiments in silico and as well as in small animals.

Briefly and clearly expressed aim of the work, the bibliography on the topic.

Unfortunately, the quality of photos and diagrams in print is poor (office printer HP LaserJet). It is necessary to consider the position of the figures and captions in relation to the text as well.

Title: If there is a word “fast” in the title, time of action/imaging should be discussed. I did not find this issue in the discussion.

Use of incomprehensible terminology (some examples only):

·       Line 17: “ suppressed” in relation to lateral resolution,

·       Line 47: “defect” - in relation to B-mode method,

·       Many words “schematic” in technical contexts instead of diagram, scheme etc.

·       Line 118: “Iris” - according to the dictionary it is a flat, colored, ring-shaped membrane behind the cornea of the eye, with an adjustable circular opening (pupil) in the center. Is it about iris diaphragm?

Figure 1a and Figure 1b - both parts are needed but their location in the text makes them small and illegible. Please enlarge or sharpen drawing 2b if possible.

Line 122: “the ANSI safety limit“ – missing reference to literature.

Line 160: is MATLAB code available in Supplementary Materials?

Section 2.4: Chicken breast muscle is usually considered to be a relatively homogeneous and has therefore been widely used for studies. Physical properties (optical, mechanical) depend on the direction of the muscle fibers. How was the phantom cut out?

Line 169: albumin and ICG solution in what solvent? Why were these "ingredients" used to make the phantom?

Section 2.5: How many rabbits were used in the experiment? 1, 2 or more?

Line 271: “left hip” means where? A vessel, muscle, bone? ICG in what solvent?

Reviewer 2 Report

1) The title suggests that it is a fast algorithm. However, there is no evidence in the paper about the reconstruction speed. While there are real-time algorithms an algorithm taking 8 seconds to reconstruct can not be considered fast.

2) The acoustic field of the transducer is not characterized. Although the focus zone is considered in the Gaussian assumption, the choice of parameters is not justified based on the transducer used. I think this is a major aspect the authors should look into to make the work scientifically relevant. In reality, one can make the Gaussian profile narrow to get better lateral resolution but is it really mimicking the acoustic field of the transducer is important to know.

3) Please also add scale to lateral and axial direction with units in result images. For example in Fig. 4 the scale on the lateral direction is missing. Furthur in table 2 as well the units are missing.

4) In the simulation studies please also include targets in the lateral direction. Now there are only targets along the axial direction. It is important to know when the proposed algorithm misses adjacent targets in the axial direction, especially at larger depths.

Round 2

Reviewer 2 Report

The authors have addressed all the concerns. I do not have more comments.